# Cardiomyocyte Ploidy, Metabolic Reprogramming and Heart Repair

**DOI:** 10.3390/cells12121571

**Published:** 2023-06-07

**Authors:** Andrea Elia, Sadia Mohsin, Mohsin Khan

**Affiliations:** 1Center for Metabolic Disease Research, Lewis Katz School of Medicine, Temple University, Philadelphia, PA 19140, USA; 2Cardiovascular Research Center, Lewis Katz School of Medicine, Temple University, Philadelphia, PA 19140, USA; 3Department of Cardiovascular Sciences, Lewis Katz School of Medicine, Temple University, Philadelphia, PA 19140, USA

**Keywords:** ploidy, cardiomyocyte, cell cycle, metabolism, binucleation

## Abstract

The adult heart is made up of cardiomyocytes (CMs) that maintain pump function but are unable to divide and form new myocytes in response to myocardial injury. In contrast, the developmental cardiac tissue is made up of proliferative CMs that regenerate injured myocardium. *In mammals*, CMs during development are diploid and mononucleated. In response to cardiac maturation, CMs undergo polyploidization and binucleation associated with CM functional changes. The transition from mononucleation to binucleation coincides with unique metabolic changes and shift in energy generation. Recent studies provide evidence that metabolic reprogramming promotes CM cell cycle reentry and changes in ploidy and nucleation state in the heart that together enhances cardiac structure and function after injury. This review summarizes current literature regarding changes in CM ploidy and nucleation during development, maturation and in response to cardiac injury. Importantly, how metabolism affects CM fate transition between mononucleation and binucleation and its impact on cell cycle progression, proliferation and ability to regenerate the heart will be discussed.

## 1. Introduction

The adult mammalian heart is characterized by limited capacity to repair and regenerate [1,2,3]. Gradual loss of cardiac cells, which occurs following injuries and heart disorders, results in significant cardiac maladaptive remodeling that culminates in severe heart dysfunction. Notably, the adult heart undergoes several morphological and functional alterations following detrimental stimuli, resulting in a continuous replacement of viable myocardium with activated fibroblast, responsible for scar formation and adverse cardiac hypertrophy [4]. Clinical studies show cardiac tissue derangement negatively impacts regenerative response of cardiomyocytes (CMs), unable to replace lost myocardial tissue [5,6,7]. In addition, hostile cardiac microenvironment after injury influences the survival and reparative activity of remaining CMs [8,9,10]. Similarly, adult CMs show limited self-division and cell turnover activity in response to pathological stress. Therefore, new innovative treatments are required to better understand and promote the reparative and regenerative mechanisms in the heart. Interestingly, developmental heart manifests remarkable proliferation and regeneration ability that gradually decreases in the perinatal and postnatal stages, characterized by DNA replication of the CMs, without cytokinesis, thus leading to progressive binucleation [11]. This is accompanied by increase in CM size and myocardial hypertrophy to compensate for functional demands on the cardiac tissue with organismal growth [1,12]. Hence, this suggests that CMs exhibit three different degrees of growth and cell cycle modulation; from proliferation to binucleation, until hypertrophy. Conversely, the newborn CMs represent an important proliferative source. Several studies point out that the poor turnover in the adult CMs arrested in cell cycle, represents the main limiting factor in recovering function to the injured heart [1,13]. A further alternative approach to promote the harmed cardiac tissue restoration is based on non-muscle cells fate reprogramming to convert them into cardiac cells, to replace the dead cardiomyocytes. Interestingly, this strategy exploits the gene regulatory web involved in the growth of cardiac tissue to reintroduce the signaling pathways activated during the fetal stage [14,15]. Therefore, embryonic factors reactivation such as transcription factors (e.g., GATA4, HAND2) [16,17] cell cycle modulators (such as cyclins, cyclin-dependent kinases) [1,18], proto-oncoproteins [13], and microRNAs (miR-1, miR-133, miR-208) [19,20,21], in the adult heart could stimulate the cell cycle reentry and dedifferentiation of the mature cardiomyocytes into new and “young” cardiac cells. Moreover, these fetal regulators might help to reprogram the cell cycle of non-cardiac cells stimulating their dedifferentiation in cardiomyocytes, thus repairing the damaged heart and restore cardiac homeostasis [22]. Therefore, the multiple and promising evidence about the proliferation and development modulation of the fetal heart may provide encouraging novel strategies to support the repair mechanisms of the cardiac tissue. Likewise, the current discoveries about the regulators of the embryonic cardiac cell cycle might represent the future therapeutic targets for promoting heart regeneration, offering effective alternative approaches for cardiovascular disorders treatment. Additionally, we speculate that it would be interesting to dissect the different molecular pathways involved in liver cell proliferation. Indeed, as broadly known, the hepatic organ is characterized by a rare and unique elevated reparative capability that allows regenerating until 70% of its parenchyma [23]. Hepatocytes show changes in ploidy during physiological homeostasis, regeneration, and following injury. Specifically, nuclear polyploidy (i.e., DNA yield increase per nucleus), and cellular polyploidy (i.e., enhancement in the number of nuclei per cell) actively participate in the liver polyploidization process [24,25,26,27]. Moreover, in contrast to CM polyploidization, liable to the cell cycle arrest and loss of proliferative capability [28], liver polyploidy does not impact the hepatic regenerative activity in mice, thus favoring hepatic cells’ proliferation throughout life [29]. Following failed cytokinesis murine liver cells undergo polyploidization upon weaning, increasing progressively with aging [30,31]. Interestingly, Celton-Morizur and collaborators identified the cellular signaling pathway modulating the genesis of binucleated tetraploid hepatocytes, which represents the key phase for the establishment of murine liver polyploidization. Notably, regulation of the Akt pathway modulates cytokinesis failure episodes, thus suggesting that the PI3K/Akt molecular axis, downstream from the insulin signaling, is pivotal to the tetraploidization phenomenon [32]. Polyploid hepatocytes start cell division by generating multipolar spindles, leading to the formation of both mononucleated and polyploid new cells [29]. Additionally, by using the modular biology approach and genome-scale cross-species comparison, Anatskaya et al., demonstrated that somatic polyploidy represents an evolutionary tool for quick adaptation to stress and environmental changes, promoting cell function under stress and energy depletion both in the heart and in the liver. In both compartments, polyploidy preserved cells’ vitality (primarily associated with sirtuin-mediated pathways activation), induced the ATP reserve mobilization, and enhanced tissue-specific functions through a metabolic shift from aerobic to anaerobic respiration, thus limiting cell energy impoverishment [33]. Hepatic regeneration that occurs after liver injury determines a specific selection of the most genetically resistant hepatocytes that participate in the reparative mechanisms [34,35]. Therefore, analogously to the hepatocytes, we speculate that selective disease-resistant clones can be amplified to protect the heart from degenerative mechanisms and pathological states. Moreover, hepatic cells undergo different levels of polyploidization over time from liver development to aging increasing or reducing this process after injury. Thus, it would be interesting to dissect the mechanisms and the gene rulers that trigger the physiological liver regeneration and try to target them to modulate the cardiomyocytes’ proliferation post-injury. Lastly, it would be stimulating to identify a specific target involved in liver cell regeneration- such as Notch’s molecular pathway, especially relevant in the embryonic development stage and cell homeostasis preservation [36]. So, besides the liver, most likely in the heart as well, this signaling pathway might represent a novel future promising molecular target to rule the cardiomyocytes’ cell cycle.

## 2. Cardiomyocyte Cell Cycle Dynamics during Development, Adulthood and Injury

CMs possess ability to proliferate during early postnatal stages but lose it by postnatal day 7. Before permanent cell cycle withdrawal, postnatal CMs go through a last wave of incomplete cell cycle, characterized by the uncoupling of karyokinesis from cytokinesis, thus triggering CM binucleation [37,38]. Field and colleagues have demonstrated that DNA synthesis in murine CMs during postnatal development features two specific stages [39]. The first phase takes place during the fetal period, where thymidine labeling rates of 33% for DNA synthesis were detected in mice ventricles at embryonic day 12 (E12). Notably, during this phase, karyokinesis and cytokinesis were paired, resulting in CM proliferation. Conversely, the second stage befalls in the neonatal phase characterized by a peak thymidine labeling at day 4–6 after birth. This stage is recognized as acytokinetic mitosis, and features endoreplication without cell division, thus resulting in ventricular CM polyploidy. Specifically, polyploidization, characterized by significant DNA yield increase, detected during CM development, comprises two specific phases: endoreplication, which coincides with mitosis inhibition, and multinucleation, namely cytokinesis inhibition. Various modulators drive cardiac polyploidization thus ensuring to stimulate specific cell responses following different physio-pathological conditions. Importantly, the total abolition in cardiac polyploidization might induce a severe impairment in the cell replication process that would require the entire cardiac cell cycle reprogramming because highly lethal for the heart [40]. Hence, polyploid cardiomyocytes might represent a specific checkpoint for the cell cycle, which splits cell division from DNA duplication. Next, cells undergo several DNA replication waves avoiding the mitosis phase, thus generating autopolyploid cells, which are unable to replicate anymore. Intriguingly, Gerdes et al. through in vitro models described the formation of binucleated myocytes organized in an actomyosin contractile ring, in absence of abscission, well-known as mid-bodies [38]. The accumulation of myocardial binucleated cells in rodents begins around day four, peaking at the third postnatal week, where 85–90% of binucleated ventricular cells are detected [39,41]. Importantly, these levels of binucleation slightly differ somewhat among species. In swine models, the percentage of binucleated myocardial cells can reach up to 32% [42], whereas in human estimates show binucleated CMs numbers between 25–57% [43,44]. Not surprisingly, this cell division exit is followed by a significant decline in positive cell cycle modulators and consequential up-regulation in Retinoblastoma (Rb) gene expression and cyclin-dependent kinase inhibitors (Cdkl) such as p21 and p27, similar to skeletal muscle cells [45,46,47,48]. The physiological meaning of binucleated cardiac cells is still unclear, but it has been proposed to be a compensatory response in metabolically active cells where the ability to synthesize twice the RNA for protein assembly might be beneficial.

Several studies on cardiac development show that the yield of DNA replication in the adult murine heart is a topic still debated. However, immunofluorescence analysis with tritiated thymidine or BrdU staining performed in murine heart suggest that the number of CMs entering the cell cycle in an adult heart is very low [49,50]. Notably, Field and colleagues showed through tritiated thymidine incorporation analysis the limiting density of ventricular myocytes that exhibited DNA synthesis in healthy mature mice hearts [51]. Albeit, in post-ischemic murine heart, DNA replication does not look to increase meaningfully [52], a long body of evidence demonstrates significant myocyte proliferation index in end-stage of human heart failure with mitotic rate estimates ranging from 0.015–0.08% [53,54]. Therefore, it’s reasonable to postulate that a poor intrinsic proliferative power in adult CMs remains a big factor for development of cardiac disease pathology. Interestingly, despite cell cycle reentry has been detected rarely in the adult murine CMs following stress or harmful responses, several solid findings demonstrated that it likely occurs at low levels in the adult human myocardium [53,55]. In line with these results, numerous pieces of evidence support the hypothesis of elevated DNA amount per nuclei and nuclei per CM in the human models of cardiomyopathy [56]. Nevertheless, while cell cycle reentry may be limited in the human versus murine hearts following stress and/or injuries, the final fate of these cardiac cells and whether species-specific variations exist in proliferative capacity remain to be elucidated. On the opposite, Oberpriller and coworkers have observed a significant mitotic index in mononucleated CMs with respect to binucleated cells in newt myocardial tissue [57]. Specifically, the entering of binucleated newt myocytes in the cell cycle leads to nucleated cells largely compared to the myocytes that undergo cytokinesis. Hence, the documented discrepancies in the rate of mononucleated heart cells as opposed to binucleated ones that emerged among the different species [39,41,42,43,44] might explain some variations found in the proliferative potential. However, this may be accounted for “tetraploid checkpoint” presence that eliminates the binucleated myocytes in the G1 cell cycle phase, which are not able to undergo cytokinesis [58]. Interestingly, although the human heart shows a significant population of mononucleated myocytes with a high potential of cell cycle reentry, a poor regeneration ability after cardiac injury was found. Conversely, other researchers have hinted that the key factor that might trigger adult CM proliferation is their cellular size rather than the nucleation phase. Consistent with these hypotheses, studies demonstrated that as a response to heart injuries, smaller human CMs displayed higher proliferation rate than the larger more mature cells [53,54,59]. Albeit, Beltrami suggested that a low percentage of CMs entered in mitosis in the adult injured myocardium indicated their ability to reenter in cell cycle [53], an alternative further explanation might be related to cardiac myocytes endoreplication, as Adler and colleagues have observed in the human heart with no concrete evidence concerning CM proliferation [55]. Moreover, Laguens has described those human myocardial cells show a temporary and restricted cell division after heart failure, thus resulting in an endomitosis and polyploidization of CMs [56]. Intriguingly, enhancement in endoreplication and polyploidization have also been observed in adult transgenic murine hearts [60]. In addition, endoreplication might explain the discrepancy between the reported reparative ability of the myocardium after injury and that suggested based on cardiac cell cycle impairment. Yet, no recent convincing data support the formation of an actomyosin contractile ring in the adult heart of any species, a pivotal factor for any cytokinesis [38].

## 3. Polyploidization and Nucleation

Currently, the biggest challenge in basic cardiovascular science is to eliminate the cardiomyocytes loss, found during several cardiovascular disorders and improve heart regeneration. As largely described, the CMs maintain proliferation ability during the pre-natal phase thus favoring reparative and regenerative capability of the heart following injuries. Conversely, the adult myocardium shows a very low power of regeneration unable to ensure an effective replacement of the dead cells upon damage [61]. The mammalian heart exhibits a significant enhancement in CM number during the post-natal period. This process, known as cardiac hyperplasia, is characterized by a remarkable increase in CM DNA levels, due to multinucleation and nuclear polyploidization [62,63,64]. Nonetheless, CM hyperplasia is replaced by hypertrophy, namely the increase in the cellular size as heart matures. Interestingly, some studies have demonstrated a correlation between hypertrophy, multinucleation and nuclear polyploidization [65]. However, additional molecular and regeneration studies are needed to shed light on the link between heart hypertrophy and polyploidy’. Moreover, the physiological role of polyploidy in cardiac tissue and the progressive increase in heart hypertrophy compared to hyperplasia remains still unclear. Importantly, available methodological methods for analysis of cardiac proliferation, ploidy, and multinucleation show concerns and pitfalls. Therefore, new dependable tools are urgently needed to fill this gap. Recently, time-lapse represents the only technique used to conclusively measure cell proliferation, yet is limited by ex vivo preparation or isolation of CMs [66]. Interestingly, the increased number of polynucleated and polyploid CMs resulted in limited heart regeneration [28,67] ability over the mononucleated diploid myocytes [68,69]. Moreover, the physiological meaning played by ploidy and multinucleation in the adult myocardium and their weight in cardiac diseases is still to be clarified. Curiously, as shown by different investigations, vertebrates with high ability for heart regeneration exhibit diploid cardiomyocytes [70], suggesting that polyploidy might limit CM proliferation and regeneration of the myocardial tissue following injury [71]. In contrast, other studies have demonstrated self-replication in polynucleated CMs, thus hinting at a novel role played by polyploidy in cardiac mechanisms of repair and regeneration [72]. Lastly, it’s very interesting that besides the cardiac tissue other organs, such as the liver, are impacted by the polyploidy process with a high proliferation and regeneration index [73]. Therefore, it would be intriguing to study the hepatic proliferation and regeneration pathways to better understand the molecular targets to use as new potential tools to improve heart regeneration capability. Could the liver represent the key organ to unveil the “codes” needed for cardiac regeneration?

### 3.1. Role of Ploidy/Nucleation during Development, Postnatal and Adult Phases

Evidence from literature suggests changes in polyploidy and nucleation impact postnatal CMs transition between proliferation and terminal differentiation (Figure 1).

However, the trigger signals of this endo-nuclear mechanism are still debated. Interestingly, Fink and coworkers have observed significant changes occurring in early postnatal development in murine hearts [74]. After birth, the cardiac tissue exhibits a remarkable enhancement in blood pressure resulting in gradual stress in the walls of the cardiac chambers associated with alteration in heart metabolism. This prolonged and sustained metabolic activity through time leads to a progressive reduction in cell proliferation followed by cessation of the cardiac regeneration process [75,76], with a parallel increase in binucleation and polyploidization. Notably, the loss of mammalian CM capacity to proliferate and divide after birth is a consequence of the changes in the intrauterine environment, resulting in intrauterine growth restriction (IUGR), associated with an elevated risk to develop serious disorders both in childhood and in adult life [77,78,79]. Additionally, IUGR severely affects the development of the heart in various species, significantly reducing the number of CMs at birth [80,81]. Specifically, a long body of evidence demonstrated the decrease in neonatal CMs number IUGR-mediated and how the species and/or the sex of the animals impacts IUGR actions, and likely its long-term effects [80,82,83]. Therefore, in order to evaluate the IUGR-mediated consequences on CM development, an unbiased stereological method to quantify the number of CMs and collect morphometrics information of the heart tissue was developed and widely adopted in cardiovascular research studies [84].

Besides morphology and structural adaptations, the postnatal CMs show specific electric modifications in response to physiological stimuli that occur during fate transition: from proliferation to terminal differentiation. Interestingly, Chen et al. have described a different electrical phenotype exhibited by mononucleated and binucleated CMs resident in the left atrium and pulmonary vein. Notably, the binucleated CMs showed significant arrhythmogenic activity with increased expression in gap junctions and intercalary discs thus allowing a fast propagation of the electrical impulse through the entire organ.

Additionally, ploidy also influenced calcium dynamics in CMs, increasing the expression of genes involved in calcium handling/excitation-contraction coupling pathways [85]. Lastly, a recent study documented a significant density of binucleated CMs in ventricular chambers compared to the atria in a mouse model [86]. Therefore, it would be reasonable to further investigate whether and how the metabolic and electrophysiological phenotypes observed in different types of cardiac cells may participate in the cell cycle exit and polyploidy, resulting in loss of cardiac regeneration capacity.

### 3.2. Nucleation/Ploidy after Myocardial Injury

Multinucleation and polyploidy represent typical characteristics in mammalian post- natal CMs, coinciding with terminal differentiation when most CMs become acytokinetic [87]. Additionally, myocardial ischemic injury and maladaptive cardiovascular hypertrophy found during heart disorders, promote a significant increase of polyploidy levels in mature CMs, thus indicating a key role played by polyploidization regulation of cell processes following pathophysiological stimuli [63,88,89]. In this respect, the immature myocardium of patients affected by a Tetralogy of Fallot (TF), a congenital heart defect, undergoes hemodynamic overload and hypoxemia after birth, thus resulting in a complex morphological remodeling. This culminates in a ploidy increase along with CMs size associated with ultrastructural dystrophic signs and incomplete cardiomyocytes differentiation, thus affecting cardiac proliferation [90]. Analogously, Fink et al. demonstrated how an increase in polyploidy modulates chromatin architecture and gene expression, inducing specific cell adaptations in response to physiological and pathological stimuli [74]. In mature cardiac tissue, characterized by restricted regeneration capability and limited CM proliferation, polyploidization could represent an effective alternative to counteract the growing mechanical and metabolic stress stimuli, following injury. However, molecular signaling pathways involved in the modulation of multinucleation and polyploidization remain still unclear, as well as increase in DNA amount CMs following heart disorders [91]. Additionally, chromatin reprogramming-based strategies through epigenetic modulators may represent a potentially valid approach to modulating CM regenerative mechanisms [92,93,94]. Notably, epigenetic modifications significantly impact gene expression and DNA binding proteins [95]. Accordingly, targeting epigenetic regulators in neonatal CMs, and amplifying them in adult CMs enhance the reparative mechanisms and regenerative capability of the heart. Therefore, the regulation of neonatal epigenetic mechanisms helps to improve CM regeneration in adult injured myocardium [96].

Cardiac hypertrophy and polyploidization are closely connected and essential for maintaining CM function [97]. Specifically, as emerged by several current investigations PI3K/Akt molecular axis is significantly involved in modulation of cardiovascular hypertrophy, besides participating significantly in the polyploidization control of the vascular smooth muscle cells [98] and in the formation of the binucleated tetraploid hepatic cells. Despite, the discrepancies observed in the polyploidization processes of these different cell types it is important to highlight the tight connection between the enlargement of cell size and the increase in chromosome number. Additionally, as largely demonstrated the cardiac hypertrophy induced by hypertension leads to polyploidization enhancement in the human heart [63,89,99]. Notably, Liu and colleagues have described similar mechanisms in a murine model [40]. In mammals, hypertrophic growth along with increased cardiac output and CM polyploidization, represent the main changes in the fate transition from postnatal to adult [100]. These modifications culminate in cell cycle arrest and mitochondrial maturation, resulting in regenerative capacity loss with high susceptibility to heart failure and injury in the mature myocardium [101]. Interestingly, several transcriptional modulators regulate molecular and cellular processes that occur in CM during the transitional period from neonatal to adult states [102]. Therefore, it is reasonable to hypothesize dissecting the dynamic transcriptional modulation in CMs during the post-natal stage, may suggest novel therapeutic targets to improve the regenerative potential in the adult injured heart [103].

However, besides the pathological and deleterious hypertrophic phenotype found in post-myocardial injuries, the exercise training also may promote a progressive increase in cardiac cell size thus resulting in a new physiological phenotype of heart hypertrophy, especially found in athletes [12].

Interestingly, a current investigation has described a significant enhancement in murine mature CM proliferation after exercise training accompanied by a slight increase in cardiac polyploidization [104]. Therefore, we suggest investigating the potential modulating effects mediated by the hypertrophic trained heart on the human CM polyploidy and cell cycle activity. Maybe, the exercise training might promote a “new phenotype” of heart polyploidization as observed for the physiological cardiac enlargement found in the trained myocardium. On the contrary, following injury, the greater part of the murine and human CMs become polyploid as shown by two different recent studies [56]. Thus, we speculate that the exercise training could represent the novel future physiological tool to positively modulate the CM polyploidization, especially after the harmful stimuli. Interestingly, the synergistic effect mediated by cardiovascular hypertrophy and polyploidization would seem to preserve the morphology of the heart organ, the electrical activity, and the contraction/relaxation coupling signaling pathway of the CM. Additionally, Vinogradov et al. have described a significant antiapoptotic activity modulated by polyploidization mechanism, thus promoting the viability enhancement both in cardiac and hepatic cells [105,106].

Although, aging is characterized by a significant decrease in cardiac mass this is not accompanied by a parallel polyploidy reduction [88]. Conversely, a remarkable decline in the average of polyploidy CM was observed in the human heart after implantation of the ventricular device [107,108]. However, based on several controversial investigations, whether cardiac polyploidy plays a beneficial or detrimental effect remains still unclear. Therefore, further detailed studies are needed to better understand the molecular and functional meaning of the polyploidization process, to control its inhibition or stimulation thus improving the cardiomyocytes’ activity and forestalling the cell impairment.

### 3.3. Strategies Targeting Ploidy/Nucleation and Their Effect on Myocardial Regeneration

Several reports have elucidated the possibility to regulate polyploidization mechanism in cardiac tissue, through manipulation of specific modulators involved in cell replication, thus promoting heart cell proliferation (Table 1) [105,106]. Cell cycle progression in mammals is controlled by mitogenic stimuli induced by cyclically expressed regulators, recognized as cyclins, which in turn orchestrate characteristic checkpoints during different phases of the cell cycle, through cyclin-dependent protein kinases (Cdks). CDK4 is mainly involved in the transition from the G1 to S phase of the mammalian cell cycle thus determining the initiation of DNA replication [106,109]. Moreover, together with CDK6, CDK4 acts as a key regulator of the late G1 phase [110] and is involved in the transition from G1 to S phase of the mammalian cell cycle thus determining the initiation of DNA replication. Both CDK4 and CDK6 organize in active enzymatic complexes together with cyclin D1, D2, or D3 [111] preferentially involved in Rb family members phosphorylation, thus favoring E2F release. Specifically, this transcription factor participates in G1 withdrawal and DNA duplication in mammalian cells. Interestingly, Cyclin D1 upregulation promotes CM DNA synthesis and multinucleation enhancement in the transgenic in vivo model [106]. Conversely, cyclin G1 expression, known as p53’s [112,113,114] transcriptional factor, highly participated in CM polyploidization and consequential proliferation arrest in the heart. Specifically, cyclin G1 overactivity increased significantly cardiac DNA synthesis with a parallel cytokinesis abolition, thus resulting in CM multinucleation [40]. Additionally, cyclin G1 null mice display significant decrease in CM ploidy following cardiac hypertrophy induced by pressure overload, likely attributable to apoptosis signaling abolition and cell vitality enhancement. In line with this, Levkau demonstrated the key role of survivin in CM mitosis control in a specific transgenic mouse. Notably, survivin genetic deletion severely influenced the CM nuclear shape, leading to a significant enhancement both in DNA synthesis and cardiac hypertrophy [105].

Another specific cardiac cell-cycle target to regulate polyploidization is YAP1 (Yes-associated protein). This transcriptional effector belonging to the Hippo signaling pathway is involved in heart cell proliferation, improving cardiac function [115] and cell viability in response to ischemic injury [116]. Along this line, Liu et al. using a gene strategy based on adeno-associated virus 9 (AAV9) investigated locally knocking down the Hippo signaling adaptor gene (Salvador- Sav) in border zone CMs in a myocardial post-ischemic porcine model. Interestingly, pigs treated for three months with AAV9-Sav-shRNA, which was inoculated through catheter-driven subendocardial injection, showed a sizeable improvement in cardiac function, associated with a relevant enhancement in capillary density and significant reduction of myocardial fibrosis, compared to animals treated with AAV9-GFP, as control. Additionally, AAV9-Sav-shRNA topic therapy induced a decline in CM polyploidization thus promoting cell division and cardiac renewal potential after injury [117]. Accordingly, Hippo signaling’s local silencing may represent a novel promising target to treat cardiac disorders, arresting cardiomyocytes’ polyploidization, thereby increasing the reparative and regenerative mechanisms in adult failed hearts. Similarly, Martin and colleagues demonstrated that YAP-1 acts as an endogenous repressor in adult heart renewal and regeneration [118]. Specifically, the YAP-1 null adult CMs showed a significant increase in cytokinesis levels accompanied by cell-cycle arrest. Conversely, the YAP-1 adult mutant injured CMs exhibited elevated regeneration capacity without polyploidy evidence in murine model [115].

Therefore, it’s reasonable to hypothesize that Hippo/YAP-1 signaling axis might represent a novel promising target to treat cardiometabolic disorders, through the modulation of the cardiac cell cycle and regeneration mechanism [119]. In this respect, ZEB1 (Zinc Finger E-Box Binding Homeobox 1) was identified as a crucial transcription factor entailed in mouse CM proliferation and terminal differentiation. Notably, Bak and colleagues dissected trascriptome in neonatal CMs, combining fluorescence activated cell sorting (FACS) with single cell RNA sequencing. Their findings showed ZEB1-mediated proliferative effect on CMs and a ploidy stratified transcriptomic profile of developing CMs useful to better elucidate the biomolecular underpinnings that occur during the fate transition from post-natal to mature CMs [120]. Recent studies have shown the reactivation of cardiac developmental factors in the adult heart as a novel way to promote CM cell cycle reentry and change percentages of mononucleated diploid vs binucleated polyploid cells post-injury [121]. Our recent work has identified pluripotent microRNA-294 as a potent driver of CM proliferation and cell cycle activity in the mouse heart post-injury [122]. Delivery of mi-croRNA-294 resulted in a significant increase in mononuclear CMs compared to binucleated cells in the injured heart. Similarly, RNA-binding protein LIN28a, active primarily in the developmental heart, was shown to increase CM proliferation when reactivated in the mouse adult heart post-myocardial infarction injury [123]. Interestingly, LIN28a augmentation of ejection fraction was positively correlated with the percentage of mononuclear diploid cardiomyocytes (MNDCMs) in the postnatal and adult heart after injury. Therefore, based on a solid plethora of data that demonstrates microRNAs play, a pivotal role in cardiovascular pathological mechanisms, including angiogenesis, cell proliferation, and cell death, we speculate that strategies based on targeting them and other non-coding RNAs may be useful to counteract cardiac disorders [124,125,126,127].

Defects in chromosome segregation might induce cytokinesis dysfunction, which in turn can trigger the apoptotic responses to promote carcinogenesis and can impact polyploidization in the heart. Notably, the impaired mitotic microtubule distribution is accompanied by aberrant actomyosin ring anchorage and can result in significant increase of binucleated CMs. Interestingly, a recent study described severe alterations in CM cytokinesis during development after deletion of growth arrest-specific 2 like 3 (Gas2l3) protein. Gas2l3 is a crucial cytoskeletal linker protein, involved in stabilizing and formation of the actin and microtubule network. Gas2l3 knock-out mice displayed multinucleated cardiac cells, thus suggesting its pivotal role in the heart polyploidization process [128]. Similarly, abnormalities in epithelial cell transforming-2 (Ect-2) protein expression induced significant dysfunction in cytokinesis, leading to polyploid cardiomyocytes [129]. Furthermore, an interesting recent study has explored the tight correlation between the occurrence of diploid cardiac myocytes and their high proliferation capacity in a post-ischemic murine model [129]. Specifically, the investigators have identified cardiac troponin I-interacting kinase (Tnni3k) gene, involved in CMs ploidy variation across several species [130]. Tnni3k deletion favors *MNDCMs* formation, thus supporting heart regeneration after injury [131,132]. Accordingly, these results might help to resolve some concerns about the CM proliferation and consequential heart regeneration limited by the polyploidization mechanism.

Conversely, significant phenotypic differences emerged in the dynamics of cell renewal and turnover in the human heart. Interestingly, *Yekelchyk* recently described a similar transcriptome profiling exhibited by mono- and multi-nucleated *mouse* adult ventricular myocytes. Notably, through single-cell RNA-sequencing the investigators evaluated the heterogeneity among CMs both in the physiological and pathological conditions, observing identical sets of genes expressed by individual mono- and multi-nucleated CMs [133]. Therefore, based on this study it seems that cardiac tissue polyploidization does not influence CM transcriptomic profile under physiological conditions, a possibility that needs further exploration.

## 4. Ploidy, Cell-Cycle Arrest, and Cell Death

In mammals during development, CMs are all mononucleated and diploid (MNDCMs) with strong inherent replicative potential. In fact, MNDCMs retain the embryonic/fetal gene imprinting, essential for the regeneration of the cardiac tissue, even after injury [28,70,100,134]. However, after birth following cardiac maturation, CMs undergo cell cycle arrest, and lose their proliferative power, thus becoming polyploid and multinucleated. Polyploid CMs show a tetraploid nucleus or higher ploidy degree, as numerous diploid nuclei, or multiple tetraploids nuclei. Notably, the uncoupling of karyokinesis from cytokinesis, which occurs in postnatal CMs, culminates in the endoreplication phase liable for CM polyploidization and cell cycle arrest. In humans, and rodents as well, the heart is composed of CMs with tetraploid nuclei or with multiple nuclei [28,135]. Similarly, porcine CMs show numerous rounds of endoreplication up to two months after birth, leading to wide bi- and multi-nucleation of all CMs [136]. In contrast, zebrafish mature heart exhibits high incidence of MNDCMs population with a high regeneration rate that ensures a permanent proliferative capability of cardiac tissue [71]. Based on the evidence that the constitutive ratio between mononuclear tetraploid and binucleated CMs changes through the mammalian species, and it further undergoes modifications within the same species, even across different murine genetic backgrounds [137], it is reasonable to speculate a genetic modulation of CM endoreplication cycle arrest either before or after karyokinesis phase. The differences observed in the several polyploidy levels among various species are mirrored in the different timing of mammal CMs polyploidization. While in rodents this phenomenon befalls during the first week after birth, human, pig, as well lamb cardiomyocytes begin the endoreplication round at the end of gestation, accompanied by further post-natal polyploidization [44,135,138,139]. Moreover, unlike most mammal CMs that undergo only one endoreplication cycle persisting over life, multiple endoreplication waves in mouse and rat CMs during weeks after birth has been found. Polyploidization is also involved in the pathological mechanisms that affect mature CMs, which undergo adverse multinucleation or enhanced nuclear polyploidization in pathological states [134]. Specifically, numerous cardiovascular disorders (i.e., myocardial infarction, and heart failure) are characterized by severe CM loss, resulting in a significant enhancement in polyploidization for the remaining CM population [140,141]. Thus, most likely through a stimulus coming from the injured myocardium, it is possible to speculate that CMs start a new cell cycle undergoing further endoreplication wave. Additionally, numerous and frequent endoreplication rounds might derange the chromosome number of the CMs nuclei. The resulting alteration of chromosome content along with the unbalanced chromosomic attachment to the mitotic spindles leads to spindle assembly checkpoint activation [142]. Usually, before karyokinesis, this checkpoint contributes to mitosis delay to ensure all chromosomes are suitably paired at the metaphase plate and hooked to their proper mitotic spindles. However, the prolonged arrest of the cell cycle, in some cell types, may induce cell death through a mechanism recognized as mitotic catastrophe [143]. In addition, impaired chromosomic segregation during mitosis results in unbalanced chromosome content in the nuclei of daughter cells, which in duplicating cell types may induce aneuploidy and cancer. Albeit, CMs do not highly replicate, polyploidization might result in deranged chromosomic yield during the several endoreplication rounds, thus leading to a potential aneuploidy state. Moreover, polyploid CMs undergo different endoreplication cycles, thus reactivating the cell cycle with a double chromosome number and further increasing the unbalance of chromosome content before or after the karyokinesis step. Another key factor involved in the CMs chromosome remodeling is associated with the alteration in the centrosome integrity. Usually, all types of cells show one centrosome that divides equally for each incipient daughter cell, before mitosis. Yet, loss of centrosome integrity may promote CM G0/G1 cell cycle arrest resulting in a further asymmetric allocation of chromosomes to the new daughter nuclei [144]. In aggregate, all these events result in the centrosome disassembly of daughter cells with an unbalanced chromosomic number, thus potentially providing the explanation for why most mammalian mature CMs achieve the post-mitotic stage. Additionally, the increase of the unbalanced chromosome content, as a consequence of several cell replication occurring simultaneously in numerous CMs, might lead to cell death. Specifically, this mechanism might explain the fatal phenotype observed in mice with cardiomyocyte-specific deletion of glycogen synthase kinase-3 (GSK-3) activity, essential to maintain physiological cardiac homeostasis. Thus, GSK-3 loss is incompatible with life leading to a CMs cell cycle dysregulation up to the mitotic catastrophe, resulting in a serious lethal dilated cardiomyopathy [145]. In parallel, primary mouse CMs, upregulated factors involved in the G2/M checkpoint override, favored cytokinesis but ultimately lead to cell death. However, CMs cell death was abolished by G1 checkpoint factors expression [146]. Therefore, we speculate that the application of strategies based on the activation of CMs proliferation in mature myocardium should take into consideration their potential salutary or harmful effects. Also, it is important to evaluate which specific target of the cell cycle signaling pathway they activate, the duration and intensity of their stimulation, and their possible impact on the chromosome content.

## 5. Metabolic Control of CM Ploidy

### 5.1. CM Ploidy and Metabolism during Development and Disease

Over the years, studies have shown that the energetic state of cells can impact ploidy, cell size and function that can then be utilized by cell signaling cascades to implement specific growth and functional endpoints [147,148]. In the heart, mammalian CMs are diploid and mononuclear during development. It is well established that developmental CMs operate under a specialized metabolic state characterized by increased reliance on glycolysis [121,149]. Recently, Hirose and colleagues showed that metabolic rates determined by levels of thyroid hormones are direcetly correlated to CM polyploidization [70].

Dysregulated adenosine 5′-diphosphate: adenosine 5′-triphosphate (ADP:ATP) homeostasis is linked to development of hypertrophic cardiomyopathy (HCM) [150]. Cardiac overgrowth is one of the hallmarks of HCM, epitomized by CM endoreplication and endomitosis that together result in multinucleation and polyploidization in an environment of depressed mitochondrial ATP synthesis. Hypertrophic CMs are energy deficient, unable to generate energy by mitochondrial oxidation, and able to drive anabolic processes to support cardiac overgrowth. Bischof and colleagues identified recently a centrol role played by ATP synthase in regulating endoreplication and hyperotrophy in patients [151]. Authors showed concomitant activation of AMPK and rb-E2F axis supports DNA endoreplication and pathologic growth, thereby identifying a cardiometabolic mechanism for regulation of endoreplication in diseased hearts.

### 5.2. Metabolic Reprogramming and CM Ploidy

Studies recently have implicated metabolic mechanisms to significantly participate in the polyploidization process and cell cycle regulation in the heart (Figure 2).

The main metabolic feature of proliferating CMs is enhanced dependence on glycolysis for energy generation [152]. However, during the transition from fetal to postnatal CMs, the progressive increase in oxygen production induces a gradual metabolic shift from glycolysis to oxidative phosphorylation, coinciding with CM cell cycle arrest [153]. Specifically, the CM mitochondria become more developed and utilize β-oxidation signaling pathway substrates for energy production [154]. Nevertheless, reliance on oxidative metabolism promotes significant enhancement in reactive oxidative species (ROS), that in case of stress is associated with a progressive mitochondria impairment and DNA damage, resulting in the cell cycle cessation [76,155]. In this respect, Cao and colleagues have observed an interesting increase in mouse neonatal CMs binucleation induced with increase in β-oxidation [154]. Similarly, inhibition of fatty acid oxidation signaling was associated with a remarkable decline in murine CM polyploidization [156]. A recent study reported significant increase in the regeneration capacity of the adult diploid CMs after ischemic injury in mouse in vivo model [157]. Importantly, the progressive increase in ROS production, which significantly rises after birth, induces the mitogen-activated protein kinase (p38-MAPK axis) activation, thus promoting cell cycle arrest and formation of binucleated murine CMs [158,159]. Another master regulator of the heart proliferation and polyploidization mechanisms is thyroxine. Notably, the elevated circulating levels of thyroid hormone directly correlates with cell cycle exit and CM polyploidization, accompanied by progressive impairment in cardiac regeneration ability. Conversely, thyroid hormone receptor antagonists induce a thyroxine decline, reducing mouse CM polyploidy, activating cell cycle, and thus increasing cardiac regeneration [70]. The thermogenic changes that occur in postnatal CMs coincide with a gradual increase in thyroxine levels associated with significant metabolic changes, as the stimulation of the β-oxidation is responsible for cell cycle exit, binucleation, and regeneration ability dysfunction, typically observed in the adult heart [160]. Using single-nucleus RNA sequencing, Cui and colleagues identified a small proliferative diploid CM population in the mouse heart that exhibits increased glycolytic properties but disappears as the heart matures [161] thereby implying a developmental molecular and metabolic signature in the CMs that is lost with cardiac tissue maturation. Our recent studies have shown that reactivation of developmental signaling factors in the heart leads to metabolic reprogramming of CMs that favors increased cell cycle activity and persistence of mononuclear diploid CMs that together augment myocardial repair after injury [123]. Similarly, loss of mitochondrial uncoupling protein 2 (UCP2) in the heart increases mitochondrial membrane potential and oxidative phosphorylation that is associated with reduced CM cell cycle activity and increased ploidy [162]. In line with this, a recent elegant study demonstrated the connection between CM metabolic reprogramming and regulation of the ploidy/nucleation process during the reparative mechanisms of the injured heart. Notably, by mean RNA-immunoprecipitation sequencing Rigaud et al., described the heart regeneration modulated by RNA-binding protein LIN28a through long non-coding RNA-H19, in a cardiac post-ischemic murine model [123]. Earlier reports already evidenced the beneficial and protective effects of LIN28a-mediated in injured adult hearts after the reintroduction of the embryonic stem cell miR-294 [122]. Importantly, LIN28a actively enhanced the persistence of MNDCMs in adult injured heart tissue [123]. The early postnatal heart is predominantly characterized by a sizeable population of mononuclear diploid CMs [39,100] that exhibit a significant proliferative power able to replenish the dead CMs, thus improving reparative mechanisms to resolve the cardiac injury. Yet, CMs regenerative capability runs out by postnatal day (P7) on cardiac maturation, changes in the heart homeostasis, and raised functional and metabolic demand on the heart due to individual growth. Specifically, MNDCMs [39,100] actively proliferate during development. However, in response to cardiac maturation stimuli, mononuclear diploid CMs undergo polyploidization and binucleation, associated with permanent cell cycle withdrawal and functional changes [39,135]. Interestingly, CM polyploidization progressively increases during the fate transition from post-natal to adult CM, resulting in characteristic changes in metabolic profile and a shift in energy production (from glycolysis to β-oxidation). In line with this, highly MNDCMs proliferative niches in the postnatal heart with a unique metabolic profile were characterized [161]. Nevertheless, MNDCMs during the maturation undergo polyploidization and multinucleation accompanied by a progressive metabolic shift to fatty acid oxidation, culminating in cell cycle arrest and loss of cardiac proliferative power. Therefore, mononuclear diploid CMs with a fetal gene pool highly preserved, represent an important candidate to prompt reprogramming-based strategies of CM metabolic profile to promote cell cycle activity and reparative mechanisms after cardiac injury [163,164]. Notably, reactivation of the embryonic signaling pathway in the CMs after the injury showed a significant enhancement in cardiac remodeling along with functional improvement [121]. Accordingly, fetal reprogramming of metabolic regulators involved in CM polyploidization and multinucleation represents a novel significant strategy to restore cardiac regenerative potential in adult heart tissue post- injury. Additionally, through single-nucleus RNA sequencing rare proliferative ventricular and atrial cardiomyocyte subpopulations were characterized by metabolic profile, smooth muscle cell signaling expression, and retinoic acid tolerance [165,166]. Similarly, recent studies identified small subsets of regenerative CMs based on their cell-specific transcriptomic signatures, thus revealing proliferative CMs in adult mice [167,168]. Therefore, the enhancement of MNDCMs’ gene profile or the reintroduction of the mononuclear diploid CMs fetal metabolic regulators may delay or limit the polyploidization thus improving the regenerative capacity and reparative mechanisms in adult CMs after damage. In this regard, recently Shen demonstrated the crucial role played by insulin-like growth factor 2 in neonatal cardiomyocytes’ regeneration by mean proliferation-competent MNDCMs [169]. Intriguingly, LIN28a is a key modulator of insulin-like growth factor 2 improving the survival genes translation in embryonic stem cells [170,171], thus further confirming LIN28a’s capability to favor mononuclear diploid CMs persistence. The cardiac metabolic profile gradually changes during development, shifting from the glycolytic pathway toward a more oxidative state, thus interrupting CM cell cycle activity [149,172]. Thus, current studies suggested reprogramming the cardiac metabolism through glycolysis reactivation as an innovative approach to enhance the reparative mechanisms in the damaged heart [163,173,174]. Along this line, LIN28a contributed to cell cycle activity and improved CM structure and function by reprogramming heart metabolism toward glycolysis dependent on RNA binding to lncRNA-H19 in cardiac post-ischemic injury [123]. Notably, lncRNA-H19 is significantly involved in the proliferative processes of various tumor phenotypes through pyruvate kinase muscle isozyme 2 activation, pivotal regulator of the Warburg effect and tumorigenesis [175,176]. Lastly, lncRNA-H19 promoted regeneration, reprogramming cardiac mitochondria activity, with glycolytic metabolism enhancement through LIN28a bing protein regulation [123].

Therefore, based on this evidence there is a greater need to explore how CM metabolic processes regulate polyploidization and cell cycle in the heart. This may lead to identification of a novel potential molecular target that regulates both cell proliferation and polyploidization in the heart post injury. Most likely, the study of cardiac mitochondria might represent a future cellular target to manipulate cardiac regeneration. Are the responses to control the CM cell cycle hidden in the mitochondria?

## 6. Future Direction, Challenges and Limitations

Based on a long body of evidence and convincing data, we think that innovative therapeutic approaches are needed to favor reparative mechanisms in the mature heart after injury, promoting the CM cell cycle and DNA replication. In this regard, it would be interesting to dissect the pathophysiological role played by the polyploidization process. Indeed, this mechanism could represent the future key solution to aid the biggest challenge in cardiovascular field: reprogramming CM cell cycle to promote cardiac regeneration post injury. Nevertheless, the specific biologic consequence of the increased CM polyploidization in the adult heart is still debated. Although, a large amount of experimental evidence demonstrates pro-regenerative effects of mononucleated diploid cardiac cells, in parallel other data shows that the polyploidization induces the CM cell cycle exit thus limiting their proliferation and cardiac regeneration. Interestingly, the progressive maturation of CMs is accompanied by a gradual increase in oxidative stress and ROS production with significant mitochondria deterioration and chromatin remodeling, resulting in polyploidization and cell cycle arrest. Thus, we speculate that cardiac mitochondria may modulate the chromatin architecture in CMs thus driving heart proliferation. Moreover, literature regarding hepatic cells polyploidization regulatory processes may represent another promising therapeutic strategy to identify key modulators possibly active in the CMs. Therefore, future approaches that focus on stimulation of cardiac regeneration post injury should scrupulously evaluate the impact of both polyploidization and proliferation to avoid any imbalance in the CM regeneration processes.

## Figures and Tables

**Figure 1 cells-12-01571-f001:**
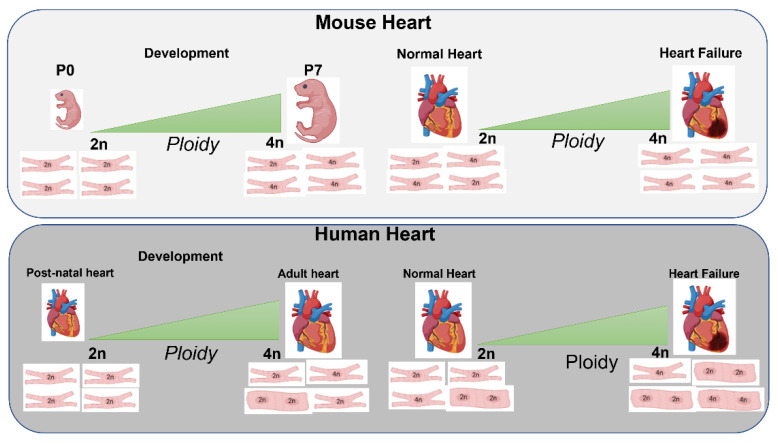
Cardiomyocyte polyploidy in relation to development and heart damage. Cardiomyocytes’ polyploidization progressively increases during the fate transition from post-natal to adult cardiomyocytes and post-myocardial injury. The accumulation of myocardial binucleated cells in rodents begins around day four, peaking at the third postnatal week, where 85–90% of binucleated ventricular cells are detected, compared to the human heart where estimates have been compassed from 25–57%. Additionally, myocardial ischemic injury and maladaptive cardiovascular hypertrophy found during heart disorders, promote a significant increase of polyploidy levels in mature cardiomyocytes, thus indicating a key role played by polyploidization in specific cell processes following pathophysiological stimuli.

**Figure 2 cells-12-01571-f002:**
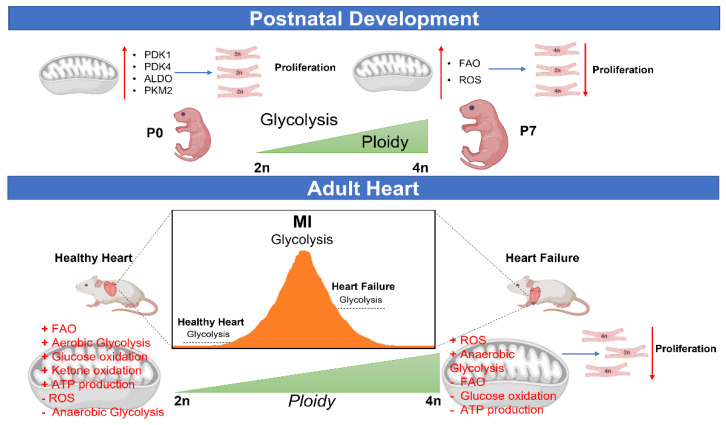
Metabolic control of polyploidization during CMs development and in the adult heart post-injury. The progressive increase in oxygen production that occurs during transition from postnatal to mature CMs induces a gradual metabolic shift from glycolysis to oxidative phosphorylation, thus determining CM cell cycle arrest and binucleation. The significant oxygen expenditure found in the ischemic heart due to impairment in metabolic debris clearance results in glucose uptake and glycolysis upregulation. Accordingly, glycolytic activity increases in post-ischemic cardiac tissue and gradually declines in heart failure accompanied by a significant enhancement in CM polyploidization levels and cell cycle arrest.

**Table 1 cells-12-01571-t001:** Strategies targeting CM cell cycle, polyploidization in the heart after injury.

Stimulus	Cellular Change	Metabolic Change	Functional Change	Signaling Pathway	References
**Cyclins D1, G1**	↑ DNA synthesis ↑ Multinucleation	Not measured	↑ Cardiac hypertrophy ↑ Apoptosis	CDK4, 6 p53	[83,86] [89,90,91]
**Hippo/YAP1**	↑ Cell cycle ↑ Cytokinesis	Not measured	↑ Cardiac function ↑ Repair and regeneration	Salv, Lats1/2	[94]
**Tnni3k**	↑ Cell cycle ↑ Diploid mononuclear	Not measured	↑ survival ↓ inflammation	Oxidative stress signaling	[101]
**Thyroid hormone**	↑ Cell cycle ↑ Diploid mononuclear	↓OxPHOS	↑ survival ↑ proliferation	Cell cycle, E2F, G2M signaling	[64]
**Gas2l3**	↑ Cytokinesis ↓ Binucleation	Not measured	↑ Proliferation ↓ Cardiac hypertrophy	p53, p21 signaling	[99]
**microRNA-294**	↑ Cell cycle ↑ Mononucleation	↑Glycolysis	↑ Cardiac function ↑ proliferation	Wee1	[97]
**LIN28a**	↑ Cell cycle ↑ Diploid mononuclear	↑Glycolysis	↑ Cardiac function ↑ Proliferation ↓ apoptosis	lncRNA-H19	[98]
**UCP2**	↑ Cell cycle ↓ Ploidy	↑Glycolysis	↑ Cardiac function	Acetyl-CoA	[115]

Abbreviations: Uncoupling protein 2 (UCP2), TNNI3 interacting kinase (Tnni3k), Growth arrest-specific protein 2-like 3 (Gas2l3), Salvador (Sav), Large tumor suppressor homolog kinase 1/2 (Lats1/2). ↑ Increased, ↓ Decreased.

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
