# Peer review of "Cardiomyocyte Ploidy, Metabolic Reprogramming and Heart Repair"

_cells, 2023, doi:10.3390/cells12121571_

Round 1

Reviewer 1 Report

In the review by Elia et al. titled “Metabolic Regulation of Cardiomyocyte Ploidy,” the authors summarize the current field of cardiomyocyte-based regeneration and repair, strategies to enhance these processes, and limitations by focusing on cell cycle regulators, signaling pathways, and alterations in metabolic properties. The review details cardiomyocyte transition from mononucleated/diploid status to polyploid/binucleated. However, my major criticism of the article is that there is a lack of emphasis on the current field of metabolic regulation of cardiomyocytes during development and disease.

My specific comments include the following:

Comment in the abstract that all cardiomyocytes are diploid and mononucleated. Please specify if this is true across all species. If the review focuses on murine models and humans, please state in the abstract and throughout the text what species you are referring to.

The description of CPCs in the introduction is not a relevant topic of the review. I would consider omitting it.

The authors focus on the role of cell cycle regulators in the strategies to target ploidy and nucleation. However, I don’t see a link describing the process of polyploidization on metabolism, as mentioned in the title of the review. The section describing the metabolic regulation of cardiomyocytes in the review is very small. How could fetal reprogramming, mentioned in the introduction, induce metabolic profiling conducive to better repair and/or regeneration?

All sections of the review should have a metabolic component, or the title of the review should be modified.

The authors mention the high regenerative potential of the liver but do not provide further details about how this is relevant to improving cardiomyocyte-based repair.

In support of modern studies that address metabolic control of cardiomyocytes, including new articles that discuss the spatial-temporal mechanisms found in cardiomyocytes following cardiac stress using single nuclei RNA-sequencing or other genomics tools, could be relevant to your study.

Author Response

We thank the reviewer for the constructive comments and response to critique as follows.

However, my major criticism of the article is that there is a lack of emphasis on the current field of metabolic regulation of cardiomyocytes during development and disease.

We thank the reviewer for allowing us to clarify. The main aim of the review is to summarize current literature on cardiomyocyte ploidy during development, maturation and disease including the impact of cardiomyocyte replenishment strategies. Cardiac metabolism has emerged recently as a target to promote cardiomyocyte cell cycle reentry. Limited literature is available as how cardiac metabolism impacts ploidy in the heart. Nevertheless, to highlight the interaction between metabolic signaling and CM ploidy, we have now divided section 5 “Metabolic control of CM ploidy” into 2 sub sections: 5.1 “CM ploidy and metabolism during development and disease” and 5.2 “Metabolic reprogramming and CM ploidy”. We hope the reviewer will find this satisfactory.

My specific comments include the following:

  1. Comment in the abstract that all cardiomyocytes are diploid and mononucleated. Please specify if this is true across all species. If the review focuses on murine models and humans, please state in the abstract and throughout the text what species you are referring to.

We are grateful for the helpful comment. We have now addressed the concern of the reviewer and we specified the cardiomyocyte characteristics (ploidy and nucleation) for all different species in the abstract and throughout the manuscript. 

  1. The description of CPCs in the introduction is not a relevant topic of the review. I would consider omitting it.

We agree with your valuable advice, and we appreciate this precious suggestion. Therefore, we omitted the part dedicated to the cardiac progenitor cells (CPCs), and we also modified accordingly the references section.

  1. The authors focus on the role of cell cycle regulators in the strategies to target ploidy and nucleation. However, I don’t see a link describing the process of polyploidization on metabolism, as mentioned in the title of the review. The section describing the metabolic regulation of cardiomyocytes in the review is very small. How could fetal reprogramming, mentioned in the introduction, induce metabolic profiling conducive to better repair and/or regeneration?

We have expanded the section 5 “Metabolic control of CM ploidy5 “Metabolic control of CM ploidy” into 2 sub sections: 5.1 “CM ploidy and metabolism during development and disease” and 5.2 “Metabolic reprogramming and CM ploidy”. We hope the reviewer will find this satisfactory.

  1. All sections of the review should have a metabolic component, or the title of the review should be modified.

We agree with the reviewer’s input. The title of the manuscript has been modified to better elucidate the purpose of the review, as advised.

  1. The authors mention the high regenerative potential of the liver but do not provide further details about how this is relevant to improving cardiomyocyte-based repair.

We thank the reviewer for this valuable advice. Therefore, we strongly enriched the Introduction section on pages 1 to 3 (lines 74- 117) and integrated the literature as you suggested, in order to provide further details about the highly useful liver regenerative power for improving the potential cardiomyocytes' reparative function.

  1. In support of modern studies that address metabolic control of cardiomyocytes, including new articles that discuss the spatial-temporal mechanisms found in cardiomyocytes following cardiac stress using single nuclei RNA-sequencing or other genomics tools, could be relevant to your study.

We really appreciate the significant suggestion of the reviewer, and we integrated the manuscript discussing several articles that adopted single nuclei RNA-sequencing or other genomics tools to evaluate the cardiomyocytes' ploidy profile and metabolic modifications.

Reviewer 2 Report

An excellent manuscript by authors from Temple University on the metabolic regulation of cardiomyocyte ploidy. The paper is detailed and describes these relationships in a reasonably precise way.

Below is a note to the manuscript:

1. the title could be more precise to emphasize that this is a narrative review,

2. affiliation number 3 is wrong - it should be "Temple University" - please remove the comma.

3. the list of references is not prepared according to Cells requirements; please change it. 

4. references should be extended to include the following papers:

https://www.mdpi.com/2073-4409/11/1/175

https://www.mdpi.com/1422-0067/23/19/11878

https://www.mdpi.com/1422-0067/22/6/3288

https://www.mdpi.com/2073-4409/11/13/2032

Author Response

We thank the reviewer for finding our article detailed and precise. We response to individual concerns as follows:

  1. the title could be more precise to emphasize that this is a narrative review,

We agree with the reviewer’s suggestion. We replaced the title of the manuscript, and we hope to better emphasize the concept of the review, as suggested.

  1. affiliation number 3 is wrong - it should be "Temple University" - please remove the comma.

We thank the reviewer for the careful suggestion, and we corrected the mistake.

  1. the list of references is not prepared according to Cells requirements; please change it. 

We thank the reviewer for the precious advice, and we accordingly modified the reference section according to the journal requirements.

  1. references should be extended to include the following papers:

We really appreciate the meaningful suggestion of the reviewer, and we mentioned/discussed the articles suggested, integrating the text of the manuscript and accordingly we modified the reference section.

https://www.mdpi.com/2073-4409/11/1/175 (lines 298- 303)

https://www.mdpi.com/1422-0067/23/19/11878 (lines 311- 319)

https://www.mdpi.com/1422-0067/22/6/3288 (lines 330- 341)

https://www.mdpi.com/2073-4409/11/13/2032 (lines 272- 276)

Reviewer 3 Report

The topic of this review is of relevance for the scientific community and I think worth being published. However, the manuscript in its current form appears rather preliminary and not really carefully crafted, resembling more a "draft" than a final version.

The review is overall well written and does indeed add new insights to the scholarly literature with respect to previously published reviews.

The presentation and critical interpretation of results of previous studies should be improved.

A section addressing ploidy, cell-cycle arrest, and cell death should be added.

The Authors should include more current reports, provide their own critical thinking and make some relevant suggestions or conclusions.

The following pertinent reports should be mentioned/discussed:

doi: 10.1242/dev.201318.

doi: 10.1007/s00395-023-00979-2.

doi: 10.3390/cells11060983.

doi: 10.1126/scitranslmed.abd6892.

Author Response

Reviewer #3

We thank the reviewer for the constructive comments and suggestions that have been helpful in revising the manuscript. Our responses o individual comments are as follows:

  1. The presentation and critical interpretation of results of previous studies should be improved.

We have done our best to improve the overall discussion of prior work in the manuscript.

  1. A section addressing ploidy, cell-cycle arrest, and cell death should be added.

We have added an additional section addressing “ploidy, cell-cycle arrest, and cell death” (lines 481- 552).

  1. The Authors should include more current reports, provide their own critical thinking and make some relevant suggestions or conclusions.

We have expanded several sections in the manuscript to provide more current and in-depth analysis of the topic. We hope the reviewer will find this satisfactory.

  1. The following pertinent reports should be mentioned/discussed:

We have added references, as suggested.

doi: 10.1242/dev.201318. (lines 497- 501)

doi: 10.1007/s00395-023-00979-2. (lines 427- 434)

doi: 10.3390/cells11060983. (lines 445- 449)

doi: 10.1126/scitranslmed.abd6892. (lines 409- 420)

Round 2

Reviewer 3 Report

Great job